# BiVWAC: Improving deep reinforcement learning algorithms using Bias-Variance Weighted Actor-Critic

## Abstract

We introduce **Bi**as-**V**ariance **W**eighted **A**ctor **C**ritic (**BiVWAC**), a modification scheme for actor-critic algorithms allowing control over the bias-variance weighting in the critic. In actor-critic algorithms, the critic loss is the Mean Squared Error (MSE). The MSE may be decomposed in terms of bias and variance. Based on this decomposition, BiVWAC constructs a new critic loss, through a hyperparameter $\alpha$, to weigh bias vs variance. MSE and Actor with Variance Estimated Critic (AVEC, which only considers the variance in the MSE decomposition) are special cases of this weighting for $\alpha = 0.5$ and $\alpha = 0$ respectively. We demonstrate the theoretical consistency of our new critic loss and measure its performance on a set of tasks. We also study value estimation and gradient estimation capabilities of BiVWAC to understand the means by which BiVWAC impacts performance. We show experimentally that the MSE is suboptimal as a critic loss when compared to other $\alpha$ values. We equip SAC and PPO with the BiVWAC loss to obtain BiVWAC-SAC and BiVWAC-PPO and we propose a safe $\alpha$ value, $\alpha^*$, for which BiVWAC-SAC is better than or equal to SAC in all studied tasks but one in terms of policy performance. We also point out that BiVWAC introduces minimal changes to the algorithms and virtually no additional computational cost. In addition we also present a method to compare the impact of critic modifications between algorithms in a sound manner.

## 1 Introduction

Most of current learning algorithms are based on the concept of empirical risk minimization (ERM). Introduced by Vapnik (1998) the empirical risk minimizer is defined in a supervised learning problem as a function that minimizes the empirical risk. In our case this will be the mean squared error (MSE):

$$\widehat{f} \in \arg\min_{f \in \mathcal{F}} \frac{1}{n} \sum_{i=1}^{n} (f(X_i) - Y_i)^2$$

where $(X_1, Y_1), \ldots, (X_n, Y_n) \in \mathbb{R}^d \times \mathbb{R}$ are the data, $Y$ is the target we want to learn, $X$ the features, and $\mathcal{F}$ is some class of functions (e.g. a neural network). This is meant as an approximation of solving the problem:

$$f^* \in \arg\min_{f \in \mathcal{F}} \mathbb{E}[(f(X) - Y)^2]$$

because we only have access to a sample of data and we do not know the distribution of $X$ and $Y$.

**ERM for Deep RL value prediction**   Deep reinforcement learning (Deep RL) is no exception to this rule. However the setting varies slightly, which leads to important considerations. In Deep RL one wishes to learn the value function $V$ (or $Q$ the state-action value function). We use supervised learning to do so through the learning of a parameterized function $f_\phi$ (most likely a neural network with weights $\phi$). However, the targets $Y$ ($V$ or $Q$) are unknown in RL. Instead a proxy $\hat{Y}$ ($\hat{V}$ or $\hat{Q}$) of those values is used. This proxy is often an estimation of the "true" values, built using the reward received at the current step $r_t$ and some combination of the estimations of the value of the next states using $f_\phi$. This differs from the classical supervised setting as, adding to the unknown targets, the proxy targets are functions of the estimator they are used to train, which leads to a non-stationary

learning problem. One may argue that this setting is not a proper supervised learning one where the "targets" (the proxy targets) are already known before prediction. However, the implicit assumption made in Deep RL is that these proxy targets $\hat{Y}$ are good enough representations of the true targets in order for $f_\phi$ to still learn about $Y$. Thus it is considered safe to assume that, through the use of a proxy, we are still learning information about the true targets. However this entails an additional error which encapsulates how good of an approximation of $Y$, $\hat{Y}$ is. This composition of error can be understood as trying to learn $f_\phi$ to reduce $MSE(f_\phi, \hat{Y})$ while also reducing the difference between $MSE(f_\phi, \hat{Y})$ and $MSE(f_\phi, Y)$ :

$$MSE(f_\phi, Y) = MSE(f_\phi, \widehat{Y}) + (MSE(f_\phi, Y) - MSE(f_\phi, \widehat{Y}))$$

The validity of the proximity between $\hat{Y}$ and $Y$ is a common belief, and the experimental validation of this assumption is nonetheless rarely studied as pointed out by Ilyas et al. (2020).

The common way to solve this supervised learning problem in RL is to minimize $MSE(f_\phi, \widehat{Y})$ to hopefully also reduce $MSE(f_\phi, Y)$. Then, we can apply the bias-variance decomposition of the MSE (James et al., 2023, Section 2.2.2) to MSE($f_\phi$, $\hat{Y}$) (although strictly speaking we are not dealing with bias and variance but bias (or variance) of an estimation of the target). This allows us to study bias-variance weightings in our prediction problem.

**Bias-variance weighting in Deep-RL value prediction**  In this work we will study how bias-variance weightings in the critic loss impacts both the critic's performance (in terms of value estimation) and the agent's performance (in terms of policy returns). Our intuition comes from the fact that the traditional critic loss (the MSE) weighs equally bias and variance, and that brings two issues: first, this 50-50 weighting is arbitrary; second, Tucker et al. (2018), Ilyas et al. (2020) and Flet-Berliac et al. (2021) argue that the core problem in value estimation is the variance and not the bias. As a consequence, we want to study how changing this weighting impacts performances, and try to understand through which aspect of the actor-critic framework the modifications happens. Understanding more about how this weighting impacts learning will allow to more efficiently select the weighting to get better results across multiple tasks, as well as giving a better understanding on how the critic impacts the learning of actor-critic algorithms.

**Related works**  Other works have studied core concepts of actor-critics algorihms and challenging the common beliefs around them. Notably, CrossQ (Bhatt et al., 2024) challenges the need for a critic in the first place, Ilyas et al. (2020) measure experimentally the validity of the assumption that the critic fits the true values. The same is also true for common building blocks of RL algorithms such as $n$-steps returns (Daley et al., 2024). There is also an important amount of work regarding improving bias and variance in RL such as Averaged-DQN (Anschel et al., 2017) or studying their properties (Zhang et al., 2021). Our work lies ate the intersections of these questions.

## Contributions

1. We introduce **Bi**as-**V**ariance **W**eighted **A**ctor **C**ritic(BiVWAC), a new actor-critic algorithms modification allowing to control the bias-variance weighting of the critic loss through a new hyperparameter: $\alpha \in [0, 1)$ and prove that the BiVWAC objective still leads to an unbiased estimation of the policy gradient $\nabla J, \forall \alpha \in [0, 1)$
2. We empirically show that BiVWAC-SAC is strictly better than SAC on all the tasks we studied using the same value for $\alpha$.
3. We empirically study BiVWAC-PPO and show that the modification can lead to better or worse results depending on the $\alpha$ value, pointing to a possible performance increase if $\alpha$ is tuned correctly.
4. We provide intuitions about the underlying mechanisms that lead to variations in performances due to BiVWAC and we provide experimental results to study those claims.

## 2 PRELIMINARIES

### 2.1 BACKGROUND AND NOTATIONS

Reinforcement Learning (RL) is solving a Markov Decision Problem (MDP). In this work we consider infinite-horizon MDPs with continuous states $s \in \mathcal{S}$ and continuous-actions $a \in \mathcal{A}$, with $\mathcal{S} \subseteq \mathbb{R}^{\|\mathcal{S}\|}$ the state space and $\mathcal{A} \subseteq \mathbb{R}^{\|\mathcal{A}\|}$ the action space, a transition function $P(s_{t+1}|s_t, a_t) : \mathcal{S}^2 \times \mathcal{A} \to [0, 1]$ (with a slight abuse of notation as we denote probabilities as a function, as in Sutton & Barto (2020)) and a reward function $R(s_t, a_t, s_{t+1}) : \mathcal{S} \times \mathcal{A} \times \mathcal{S} \to \mathcal{R}$, with $\mathcal{R} \subset \mathbb{R}$ the finite set of possible rewards. $\pi_\theta(a|s) : \mathcal{S} \times \mathbb{R}^{\|\theta\|} \to \mathcal{A}$ denotes a stochastic policy parameterized by $\theta$ $(\pi_\theta(a|s) = \pi(a|s, \theta))$. In this work we limit our scope to policies which can be represented by Gaussian distributions (where we learn the mean and standard deviation of Gaussian distributions which are then used to sample continuous actions). The agent repeatedly interacts with the environment by sampling actions $a_t \sim \pi_\theta(.|s_t)$, and observing rewards $r_t = R(s_t, a_t, s_{t+1})$ and new states $s_{t+1} \sim P(.|s_t, a_t)$. The objective is to find a policy $\pi_\theta$ that maximizes the expected sum of discounted rewards: $J(\pi_\theta) \triangleq \mathbb{E}_{\tau \sim \pi_\theta} [\sum_{t=0}^{\infty} \gamma^t r_{t+1}]$, where $\gamma \in [0, 1)$ is the discount factor. Throughout this paper $\mathbb{E}_{\tau \sim \pi}[X]$ denotes the expectation of $X$ under samples from trajectory $\tau$ generated by $\pi$ in the current task (which is considered implicit, hence the lack of inclusion of $P$, $R$ or $s_0$), this means that $a_t$, $s_t$, and $r_t$ are taken from $\tau = (s_0, a_0, r_0, s_1, a_1, r_1, ...)$, a trajectory sampled from the environment using $\pi_\theta$ through repeated sampling of $a_t \sim \pi_\theta(.|s_t)$, $s_{t+1} \sim P(.|a_t, s_t)$ and $r_t = R(s_t, a_t, s_{t+1}), \forall t \in \mathbb{N}^+$. We denote the value of a state $s$ under policy $\pi$ as $V^\pi(s) \triangleq \mathbb{E}_{\tau \sim \pi} \left[ \sum_{k=0}^{\infty} \gamma^k r_{t+k+1} | s_t = s \right]$ and the value of an action $a$ in state $s$ under policy $\pi$ as $Q^\pi(s, a) \triangleq \mathbb{E}_{\tau \sim \pi} [\sum_{k=0}^{\infty} \gamma^t r_{t+k+1} | s_t = s, a_t = a]$.

In this work we consider deep reinforcement learning where the policy and the value function are learned using parameterized function estimators (usually a neural network, hence the name). We denote $\theta \in \mathbb{R}^n$ the policy parameters and $\phi \in \mathbb{R}^m$ the value parameters.

### 2.2 BIAS AND VARIANCE

Bias-variance decomposition is a classical property of the MSE in statistics. Let us first express a general version of the decomposition of the MSE between an estimator $\widehat{y}$ and its (possibly random) target $y$. The proof of this lemma is given in Section A.1.

**Lemma 2.1** (Bias variance decomposition). *Let $y, \widehat{y}$ two random real variables. Then,*

$$\mathrm{MSE}(\widehat{y}, y) = \mathrm{Var}(\widehat{y}) + \mathrm{Bias}(\widehat{y}, y) - 2\mathrm{Covar}(\widehat{y}, y) + \mathrm{Var}(y).$$

*In particular, if $\widehat{y}$ and $y$ are independent, we recover the usual bias-variance decomposition*

$$\mathrm{MSE}(\widehat{y}, y) = \mathrm{Var}(\widehat{y}) + \mathrm{Bias}(\widehat{y}, y).$$

Bias and variance are often discussed in RL, the common consensus being that it is preferable to decrease variance at the expense of more bias, as the former is much more frequently greater than the latter, thus giving an overall improvement by decreasing the total error. However, it is rarely explicitly defined what bias and variance are referring to in this context: to which $\widehat{y}$ and $y$ should we apply Lemma 2.1? There are three main possibilities for bias and variance in deep actor-critic algorithms, coming from three different estimations (the others are beyond the scope of this paper):

1. Estimation of the empirical surrogate "true" value $\hat{Q}^\pi$ (or $\hat{V}^\pi$) using $f_\phi$. This is the classical bias and variance of a regression model.
2. Estimation of the true value $Q^\pi$ (or $V^\pi$) using the empirical surrogate "true" value $\hat{Q}^\pi$ (or $\hat{V}^\pi$). This is the bias and variance of the proxy target we use compared to the true target as we do not have practical access to the true target.
3. Estimation of the true gradient $\nabla_\theta J(\pi_\theta)$ using the policy gradient w.r.t. the policy parameters $\hat{\nabla}_\theta J(\pi_\theta)$. As we only estimate the gradients from a limited number of samples, it is only an approximation of what the gradient would be with the knowledge about the whole $J$ (which would give the direction towards the global optimum).

We argue that the more meaningful estimation to consider is the one that is the closest to the objective we are trying to solve. As the policy gradient $\nabla_\theta J$ directly reflects our objective of maximizing $J$,

it is the best candidate. The value estimation (or in other words, the critic) is only here to provide a baseline to help the actor achieve its goal. It is an auxiliary objective. There is no guarantee that reducing the variance or bias of the critic will improve the agent performance, and also that a variance and bias increase cannot lead to an improvement in agent performance. The common belief is however that the better the critic (lower prediction error), the better the performance of the agent.

To clarify notations, we are going to define the relevant bias and variances for this paper. We first recall the definition of the MSE, Bias and Variance in Statistical Learning: $MSE(\widehat{f}) = \mathbb{E}_{D,X}[(\widehat{f}_{\mathcal{D}}(X) - y(X))^2]$, where $D$ is the dataset used to train $f$, $X$ is a set of examples with $y(X)$ their labels. In our scenario, $D$ is obtained by way of a trajectory $\tau$ sampled using $\pi$. $X$ would be $\tau'$, a different trajectory sampled from $\pi$, and $y(X)$ would be some oracle predicting the desired value (e.g. $\nabla_\theta J(\pi_\theta)$). However, in practice, during the training of actor-critic algorithms, the test set $X$ is the same as the training set $D$: the sampled states and actions are first used as a test set (the agent predicts actions and values), and then the agent learns from this same set (we compute the losses based and train the network). It is possible, during or after training to consider a proper different test set to evaluate the agent. However, for this work we will only consider a shared train-test-set : $D = X = \tau \sim \pi$. Using these definitions we derive the bias and variance for the critic value estimation we get the following bias and variance ($Q$ can be exchanged with $V$ without loss of generality):

$$Bias_{critic} \triangleq \mathbb{E}_{\tau \sim \pi_\theta}\left[f_\phi - Q^{\pi_\theta}\right], \quad Var_{critic} \triangleq \mathbb{E}_{\tau \sim \pi_\theta}\left[(f_\phi - \mathbb{E}_{\tau \sim \pi_\theta}\left[f_\phi\right])^2\right] \tag{1}$$

We use $\widehat{Bias}_{critic}$ in place of $Bias_{critic}$ in our algorithms as we do not have access to $Q$. However, in the analysis part of our work we estimate both $\widehat{Bias}_{critic}$ and $Bias_{critic}$ to measure how much we are influencing both metrics with our algorithm modifications. Note that $Variance_{critic}$ is independent of the targets so $Var_{critic} = \widehat{Var}_{critic}$. Also note that strictly speaking, $\widehat{Bias}_{critic}$ is not a bias, as $f_\phi$ is an estimator of $Q$, not $\hat{Q}$ which is only a proxy.

$$\widehat{Bias}_{critic} \triangleq \mathbb{E}_{\tau \sim \pi_\theta}\left[f_\phi - \hat{Q}^{\pi_\theta}\right], \quad \widehat{Var}_{critic} = Var_{critic} \tag{2}$$

## 3 METHOD: BIAS-VARIANCE-WEIGHTED ACTOR-CRITIC

In this section, $\hat{Q}^\pi(s, a)$ and $f_\phi(s, a)$ can be exchanged with $\hat{V}^\pi(s)$ and $f_\phi(s)$ without loss of generality.

### 3.1 MOTIVATION: IMPROVING ON AVEC AND MSE AS CRITIC LOSSES

This work extends AVEC (Flet-Berliac et al., 2021) which considers the variance of the residual errors between critic output and empirical surrogate "true" values (see equation 3) as a critic loss in place of the MSE (Mean Squared Error) (see equation 4) between the critic output and empirical surrogate "true" values:

$$\mathcal{L}_{\text{AVEC}} \triangleq \mathbb{E}_\pi\left[\left(f_\phi(s, a) - \hat{Q}^\pi(s, a) - \mathbb{E}_\pi\left[f_\phi(s, a) + \hat{Q}^\pi(s, a)\right]\right)^2\right] \tag{3}$$

$$\mathcal{L}_{\text{critic}} = \mathbb{E}_\pi\left[\left(f_\phi(s_t, a_t) - \hat{Q}^\pi(s_t, a_t)\right)^2\right] \tag{4}$$

with $f_\phi(s, a)$ the critic output for state-action pair $(s, a)$ and parameter $\phi$, $\hat{Q}^\pi(s, a)$ the empirical "surrogate" estimation of the true value of state-action pair $(s, a)$. The empirical estimations of the true values depend on the algorithm being modified (i.e. Generalized Advantage Estimation (GAE (Schulman et al., 2018)) for PPO (Schulman et al., 2017), $\hat{Q}^\pi(s_t, a_t) = r_t + \gamma \mathbb{E}_{s_{t+1} \sim \mathcal{P}}\left[V_{\bar{\phi}}(s_{t+1})\right]$ for SAC (Haarnoja et al., 2018)).

Tucker et al. (2018), Ilyas et al. (2020) and Flet-Berliac et al. (2021) assert that the core problem in true value estimation is the approximation error $(f_\phi - Q^\pi)$ and not the estimation error $(f_\phi - \hat{Q}^\pi)$. In other words, they argue that the estimator $(f_\phi)$ is appropriately fitting the empirical surrogate true

values ($\hat{Q}^\pi$), however these values are bad estimates of the true values ($Q^\pi$). In order to solve this issue, Flet-Berliac et al. (2021) propose to consider the relative values: the value of states relative to their mean value rather than their absolute values in the loss to better approximate the value function. This can be seen as a similar idea of using advantages $A^\pi(s,a) = Q^\pi(s,a) - V^\pi(s)$ instead of action-values $Q^\pi(s,a)$. Flet-Berliac et al. (2021) argue that since the variance of the critic is the main problem, it should be reduced in priority at the expense of slightly increasing the bias. Their intuition is that the critic bias is already large enough so that increasing it while greatly decreasing variance will still reduce the total MSE. This leads to consider only the variance of the residual errors in their loss instead of the MSE.

We apply Lemma 2.1 to our problem, with $y = \hat{Q}^\pi(s_t, a_t)$ and $\hat{y} = f_\phi(s_t, a_t)$. Remark that $y = \hat{Q}^\pi(s_t, a_t)$ is a function of $\hat{y} = f_\phi(s_t, a_t)$ (usually Temporal-Difference-$\lambda$ (TD($\lambda$)) approximation or a Generalized Advantage Estimation (Schulman et al., 2018)) and is not independent of $y$, so we have to keep the covariance term:

$$MSE(y, \hat{y}) = Bias(\hat{y}, y)^2 + Var(\hat{y}) - 2Covar(\hat{y}, y) + Var(y).$$

The method we propose aims at improving the capability of the critic to better fit the true value function (as opposed to the empirical surrogate one) and, as a consequence, allow for better actor performances. We share the same intuitions as Flet-Berliac et al. (2021) on how to improve the critic. We think that the relative differences between value estimations are more important than the absolute differences between them, in order to learn a good critic (one that leads to better actor performances). We also think that, as opposed to classical supervised learning, the critic should learn to quickly adapt to outliers (instead of avoiding them) as they represent valuable new pieces of information (may it be positive or negative). This may be less desirable if $P$, the transition function, is stochastic and has a large variance as this would lead to capturing variance in the transition function instead of the variance due to new behavior from the policy (e.g. reaching new and rewarding states) . How large the variance has to be to be problematic is beyond the scope of this work, we will assume that the variance due to stochastic state transitions is small compared to the variance of the Q-values (the variance of the imapct of actions should be higher than the one due to the stochasticity of the environment). Finally, we want to go beyond AVEC and study the intermediate weightings of bias and variance.

We introduce a new hyperparameter $\alpha \in [0, 1)$ to weigh the variance w.r.t. the bias in the critic MSE:

$$\text{MSE}_\alpha(y, \hat{y}) \triangleq \alpha Bias(\hat{y}, y)^2 + (1 - \alpha)\left(Var(\hat{y}) - 2Covar(\hat{y}, y) + Var(y)\right) \quad (5)$$

It is important to note that while we call it $\text{MSE}_\alpha$, when $\alpha \neq 0.5$, $\text{MSE}_\alpha$ is not equivalent to the MSE. We are just using the MSE bias-variance decomposition as a basis for a new loss. This flexibility allows adapting this hyperparameter to the environment and to improve the critic capability of quickly and accurately fitting the true value function. Our intuition is that in some tasks, the variance is indeed the true problem and thus should be focused on. However in other tasks a trade-off between the variance and the bias may be more adequate. In other words, each task has a different bias-variance weighting to consider in order to attain optimal performance.

## 3.2 BiVWAC

We propose the BiVWAC loss, a parameterized weighting of $\widehat{Bias}$ and $\widehat{Var}$ of the MSE of the critic's residual errors. Setting $\hat{y} = f_\phi(s,a) - \hat{Q}^\pi(s,a)$ and $y = 0$ (which are indepedent) in equation 5 we get:

$$\mathcal{L}_{\text{BiVWAC}}(\alpha) \triangleq \alpha \mathbb{E}_\pi \left[ f_\phi(s,a) - \hat{Q}^\pi(s,a) \right]^2$$
$$+ (1 - \alpha)\mathbb{E}_\pi \left[ \left( \left(f_\phi(s,a) - \hat{Q}^\pi(s,a) - \mathbb{E}_\pi \left[ f_\phi(s,a) - \hat{Q}^\pi(s,a) \right]\right) \right)^2 \right], \quad (6)$$

with $\alpha \in [0, 1)$ the bias-variance weighting parameter, $f_\phi$ the output of the critic network, and $\hat{Q}^\pi(s,a)$ the empirical Q-function target. Setting $\alpha = 0$ is equivalent to AVEC as we only consider the variance of the residual errors. $\alpha = 0.5$ returns the MSE (scaled by $\frac{1}{2}$, which is not relevant for scale-insensitive optimizers like ADAM (Kingma & Ba, 2015)). Setting $\alpha = 1.0$ is theoretically possible but prevents us from doing meaningful bias-correction as presented in eq. 8 below. Moreover we argue that it is a not a desirable objective as it removes the connection between the loss and $\nabla J$ altogether (see Appendix B).

As we choose to consider relative errors through the use of residual errors, one may ask if the same weighting of bias-variance would also work on the "classical" MSE (Setting $\hat{y} = f_\phi(s, a)$ and $y = \hat{Q}^\pi(s, a)$ in equation 5). However we show (in Appendix A.2) the following lemma linking this formulation to the residual error one (Setting $\hat{y} = f_\phi(s, a) - \hat{Q}^\pi(s, a)$ and $y = 0$)

**Lemma 3.1.** *For any $\alpha \in [0, 1]$, we have*

$$MSE_\alpha(y, \hat{y}) = MSE_\alpha(y - \hat{y}, 0). \tag{7}$$

Since the two formulations are equivalent, we can use either one. The residual error formulation $MSE_\alpha(y - \hat{y}, 0)$ allows for more compact equations and for easier computations as we remove the need to compute covariance. This also removes the need to consider different weightings for the different variance terms by combining $\mathrm{Var}(y)$, $\mathrm{Var}(\hat{y})$ and $\mathrm{Covar}(\hat{y}, y)$. We keep for future works the study of all the possible weightings in the bias-variance decomposition (Lemma 2.1).

We will then use the residual errors formulation for the rest of this work. This also means that using the residual errors instead of the "classical" MSE formulation has no impact if using a single weight for all the variances terms, and therefore is not a method defining trait (the traditional MSE is already considering relative errors).

From $f_\phi$, learnt through the minimization of $\mathcal{L}_{\mathrm{BiVWAC}}$ we derive our bias-corrected estimator (see Appendix B for the identification of the correction term):

$$g_\phi : \mathcal{S} \to \mathbb{R} = f_\phi + \frac{1 - 2\alpha}{1 - \alpha} \mathbb{E}_\pi \left[ (f_\phi(s, a) - \hat{Q}^\pi(s, a) \right] \tag{8}$$

with $\alpha \in [0, 1)$, and $\hat{Q}^\pi(s, a)$ the empirical estimation of $Q^\pi(s, a)$

This bias-corrected estimator, $g_\phi$ satisfies the policy gradient theorem (Sutton & Barto, 2020).

**Theorem 3.2.** *If $g_\phi$ is constructed using equation 8 and satisfies the parameterization assumption (Sutton & Barto, 2020), then for any policy $\pi$ parameterized by parameter $\theta$ we have $\nabla_\theta J = \mathbb{E}_\pi [\nabla_\theta \log(\pi_\theta) Q^\pi(s, a)] = \mathbb{E}_\pi [\nabla_\theta \log(\pi_\theta) g_\phi(s, a)]$.*

See Appendix B for a proof of Theorem 3.2. In other words, $g_\phi$ can be used in place of $Q^\pi$ to estimate $\nabla_\theta J = \mathbb{E}_{\pi_\theta} [\nabla_\theta \log \pi_\theta Q^{\pi_\theta}]$. We can think of $g_\phi$ as a sort of "unbiased" estimator of $Q^\pi$ (although strictly speaking we cannot say it is).

**Corollary 3.2.1.** *If $f_\phi$ is a function parameterized by $\phi$ and trained through minimization of equation 6, then for any policy $\pi$ parameterized by parameter $\theta$, $\mathbb{E}_\pi [\nabla_\theta \log(\pi_\theta) f_\phi(s, a)]$ is a biased estimate of $\nabla_\theta J$ and its bias is equal to $\frac{1 - 2\alpha}{1 - \alpha} \mathbb{E}_\pi \left[ (f_\phi(s, a) - \hat{Q}^\pi(s, a) \right]$.*

While $f_\phi$ leads to a biased estimator of the policy gradient, its bias can still be relatively low as the critic $f_\phi$ tends to fit the empirical targets $\hat{Q}^\pi$ closely leading to a low $\mathbb{E}_{(s, a) \sim \pi} [\nabla_\theta \log(\pi_\theta) f_\phi(s, a)]$. As a consequence, $f_\phi$ can be used for a biased estimation of the policy gradient. However, in practice, as we will need to estimate the expected value of the residual error using the empirical mean of the residual errors, using $f_\phi$ instead of $g_\phi$ could allow to reduce the variance of the policy gradient estimate at the expense of the bias $f_\phi - g_\phi$. We will study this bias-variance variation in Section 4.3. We have shown that using the BiVWAC loss to train the critic, and using its bias-corrected output satisfies the policy gradient theorem. Therefore we can safely integrate it in an algorithm without loosing the policy gradient properties.

We derive the following algorithm modification scheme:

1. We change the critic loss to $\mathcal{L}_{\mathrm{BiVWAC}}$

2. We compute the correction $\delta_{\mathrm{BiVWAC}}$

3. We replace the use of $f_\phi$, the critic output (also used in $\hat{Q}^\pi$, or $\hat{V}^\pi$), with $g_\phi$ derived from $f_\phi$ and $\delta_{\mathrm{BiVWAC}}$.

These modifications are minimal and easy to introduce into any actor-critic algorithm. To evaluate the impact of our modification scheme, we will apply these modifications to two popular deep reinforcement learning algorithms, one on-policy algorithm PPO (Schulman et al., 2017), and one

off-policy algorithm SAC (Haarnoja et al., 2018). Both are actor-critic algorithms however PPO leans more on the policy gradient side while SAC leans more on the value-based side. Since our modifications will impact the critic, we expect to see a difference in the variation of performance of PPO and SAC due to their different use of the critic.

Since we want to compute $\frac{1-2\alpha}{1-\alpha} \mathbb{E}_{\tau \sim \pi} \left[ f_\phi(s, a) - \hat{Q}^\pi(s, a) \right]$ but we do not have access to the distribution of $\tau$. We will resort to estimating it empirically by computing its mean over samples from the current batch. We remind the reader that the same method is applied in order to compute the MSE in the first place, but nevertheless, adding another empirical mean in the loss increases the dependency of batch of experiences on the result. Applying the BiVWAC modification scheme to SAC and PPO we define BiVWAC-SAC

---

**Algorithm 1** BiVWAC-SAC

1: **Input parameters** : $\beta \in [0, 1], \lambda_V \geq 0, \lambda_Q \geq 0, \lambda_\pi \geq 0, \alpha \in [0, 1)$    ▷ new hyperparameter $\alpha$
2: **Initialize** policy parameters $\theta$, value function parameters $\psi$ and $\bar{\psi}$, and Q-functions parameters $\phi_1$ and $\phi_2$
3: batch $\mathcal{D} \leftarrow \emptyset$
4: **for** each iteration **do**
5:     **for** each environment step **do**
6:         $a_t \sim \pi_\theta(s_t)$
7:         $s_{t+1} \sim \mathcal{P}(s_t, a_t)$
8:         $\mathcal{D} \leftarrow \mathcal{D} \cup \{(s_t, a_t, r_t, s_{t+1})\}$
9:     **end for**
10:     **for** each gradient step **do**
11:         sample batch $\mathcal{B}$ from $\mathcal{D}$
12:         $\delta_{\text{BiVWAC}} \leftarrow \frac{1-2\alpha}{1-\alpha} \cdot \frac{1}{|\mathcal{B}|} \cdot \sum_{t=0}^{|\mathcal{B}|} (R_t - f_\phi(a_t, s_t))$    ▷ Correction
13:         **for** $(a_t, s_t) \in \mathcal{B}$ **do**
14:             $\psi \leftarrow \psi - \lambda_V \hat{\nabla}_\psi J_V(\psi)$
15:             $\theta_i \leftarrow \theta_i - \lambda_Q \hat{\nabla}_{\theta_i} \mathcal{L}^2_{\text{BiVWAC}}(\theta_i)$ for $i \in \{1, 2\}$
16:             $g_\theta(a_t, s_t) = f_\theta(a_t, s_t) + \delta_{\text{BiVWAC}}$    ▷ Corrected value
17:             $\phi \leftarrow \phi - \lambda_\pi \hat{\nabla}_\phi J_\pi(\phi, g_\theta)$    ▷ Use $g_\phi$ instead of $f_\phi$
18:             $\bar{\psi} \leftarrow \beta\psi + (1 - \beta)\psi$
19:         **end for**
20:     **end for**
21: **end for**

---

(Algorithm 1) and BiVWAC-PPO (Appendix C).

## 4 EXPERIMENTAL STUDY

To evaluate the performance of the BiVWAC-algorithms we will use the MuJoCo (Todorov et al., 2012) tasks. These tasks revolve around locomotion in complex environments with various state and action space sizes. The goal of the selected tasks is to control the articulations of a robot in order to move forward as quickly as possible. These tasks constitute a popular benchmark as they are complex enough to differentiate between powerful algorithms like PPO or SAC. All algorithms performance were evaluated over 20 seeds while values and gradients were evaluated over 10 seeds. Unless otherwise specified the results are shown for the biased version of BiVWAC (using $f_\phi$ instead of $g_\phi$), this will be justified in section 4.2. For more details about the experimental setup see the reproducibility section and Appendix F.

### 4.1 CONTINUOUS CONTROL ON MUJOCO

In order to compare the performances of the different $\alpha$ values we follow the methodology from Agarwal et al. (2021) who promote the use of mean, median and inter-quartile mean.

Table 1: Mean ($\pm\sigma$) performance comparison between SAC, AVEC-SAC and BiVWAC-SAC.

| Environment | SAC | AVEC-SAC | BiVWAC-SAC |
|---|---|---|---|
| Ant-V4 | $3722 \pm 902$ | $4918 \pm 1027$ (32%) | **$5411 \pm 577$** (45%) |
| HalfCheetah-V4 | $9980 \pm 836$ | **$10788 \pm 667$** (8%) | $10401 \pm 860$ (4%) |
| Hopper-V4 | **$2673 \pm 529$** | $5 \pm 11$ (-100%) | $2412 \pm 696$ (-10%) |
| Humanoid-V4 | $5137 \pm 447$ | **$7513 \pm 1369$** (46%) | $6593 \pm 720$ (28%) |
| Walker2d-v4 | **$4134 \pm 811$** | $119 \pm 488$ (-97%) | $4107 \pm 1040$ (-1%) |
| Mean improvement (%) | | -22.09% | **13.50%** |

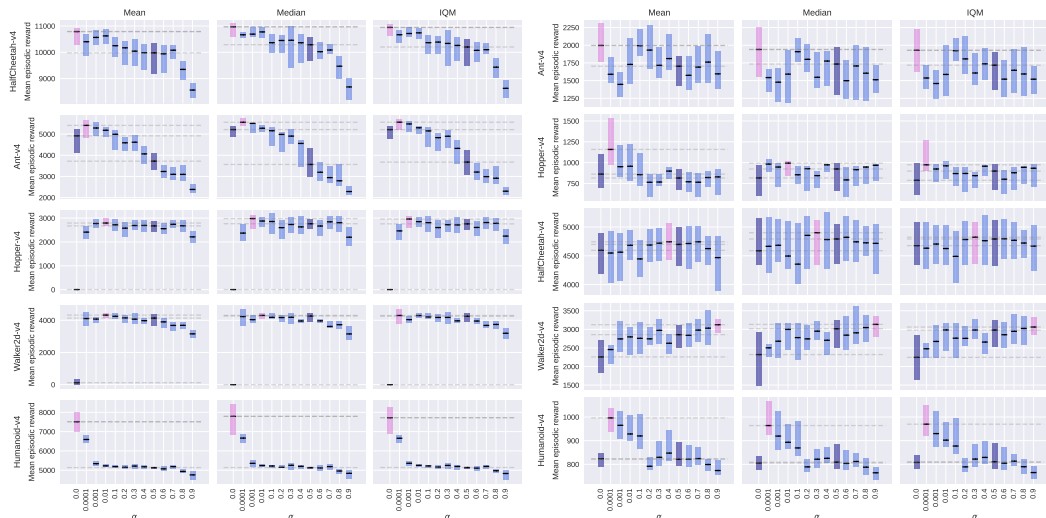

Figure 1: Average performance over the last $10^5$ training timesteps for BiVWAC-SAC (left) and BiVWAC-PPO (right) with different $\alpha$ values. Y-axis: Average episodic returns. Black lines represent the metric specified at the top of the column over 20 seeds. Bars represent 95% bootstrap confidence intervals. Pink is the best performing $\alpha$, dark blue are the baselines.

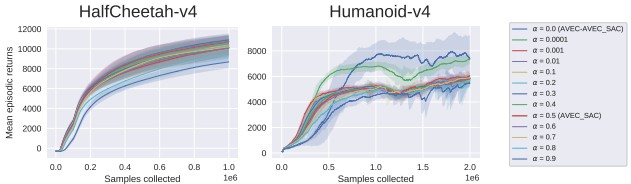

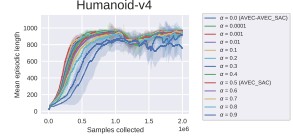

(a) Episodic mean cumulative rewards of BiVWAC-SAC for different $\alpha$ values on MuJoCo tasks. Y-axis: episodic cumulative reward. X-axis: number of collected samples.

(b) Episodic mean length of BiVWAC-SAC for different $\alpha$ values on MuJoCo tasks. Y-axis: episodic length. X-axis: number of collected samples.

Figure 2: Episodic mean cumulative rewards (left) and episodic mean length (right)

In Figure 2 we can observe that, for BiVWAC-SAC using the standard critic loss ($\alpha = 0.5$, equivalent to eq. 4) leads to suboptimal results in these tasks. Lower values of $\alpha$ outperform the MSE. The optimal $\alpha$ value varies depending on the environment. It is interesting to note that using only the variance for the critic loss ($\alpha = 0$) in BiVWAC-SAC can lead to not learning at all for(in Hopper and Walker2d) or for optimal results (HalfCheetah and Humanoid). This indicates an important aspect of not using the bias at all in the critic loss since the same cannot be said for values of $\alpha$ close to 0. As for $\alpha > 0.5$, in BiVWAC-SAC they lead to results similar worse than the MSE. We note that $\alpha = 0.0001$ leads to results better than or similar to the MSE ($\alpha = 0.5$) for all environments except Hopper, and better results than AVEC ($\alpha = 0.0$) in every environment but two (Half-Cheetah and Humanoid) (see Table 1). For BiVWAC-PPO, the improvements are less visible but that there are still $\alpha$ values performing better than the MSE. Studying the impact of $\alpha$ throughought training, we note that for BiVWAC-SAC in HalfCheetah the order of curves is almost always preserved and that in Humanoid, the best perfoming values ($\alpha = 0$ and $\alpha = 0.0001$) start with lesser performance but end up exceeding the other values after some time (See Figure 2a for BiVWAC-SAC on Humanoid and HalfCheetah, and Figure 6 and Figure 5 for the others).

Some MuJoCo environments allow for early termination of episodes if the agent is deemed unhealthy and unable to continue. This is the case for Hopper, Walker2d, Ant and Humanoid. Observing the average episodic length allows us to tell apart between policies than learnt to advance faster but failing earlier in average, with policies that advance at a slower pace but are less prone to falling. For Hopper and Walker2d the differences are marginal. However for Humanoid, while

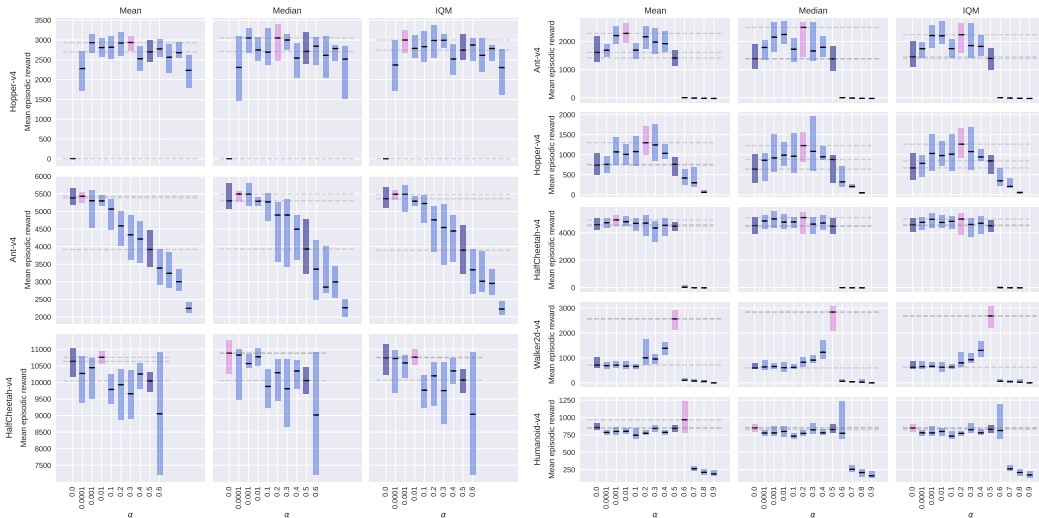

Figure 3: Average performance over the last $10^5$ training timesteps for BiVWAC-SAC with correction (left) and BiVWAC-PPO with correction (right) with different $\alpha$ values. Y-axis: Average episodic returns. Black lines represent the metric specified at the top of the column over 10 seeds. Bars represent 95% confidence intervals. Pink is the best performing $\alpha$, dark blue are the baselines.

$\alpha = 0$ leads to the best performance (see Figure 2b), it also leads to the shortest average episodic length. This means that using the AVEC loss seems to favor policies that advance quickly at the expense of robustness with regard to falling. The other episodic lengths can be seen in Appendix E.

## 4.2 CORRECTION IMPACT

As shown in corollary 3.2.1, $g_\phi$ leads to an unbiased estimation of the policy gradient, while $f_\phi$ leads to a biased estimation of the same gradient. However we pointed out that this bias may be small in practice as it is proportional to the estimation error. Meanwhile, considering the correction ($g_\phi$) may lead to more variance in the policy gradient estimation due to the estimation of the expectation of said estimation errors. To evaluate which solution is better, we study BiVWAC's performance with correction in order to conclude on whether or not it is actually helpful. In Figure 3 we can observe that the correction leads to a degradation in performance from the un-corrected version. We hypothesize that this comes from the variance introduced by the additional dependence to the trajectory samples of the estimation correction. We will thus only consider BiVWAC without correction.

## 4.3 ANALYSIS OF THE VARIANCE AND BIAS OF BiVWAC CRITIC

We study how well the critic estimates the true targets $V^{\pi_\theta}$ $or Q^{\pi_\theta}$. To do so, we measure $MSE(f_\phi, y)$ (with $y = V^{\pi_\theta}$ or $Q^{\pi_\theta}$) through its bias and variance: $Bias_{critic}(\pi_\theta, f_\phi)$, and $Var_{critic}(\pi_\theta, f_\phi)$ (see equation 1). We compare BiVWAC for different $\alpha$ values to the traditional MSE critic loss. In order to provide meaningful comparison we make sure that the critics we compare share the same initial weights, are trained on the same data, and are evaluated on the same test samples. The methodology we use is detailed in appendix **??**. To compare BiVWAC-$\alpha$ and BiVWAC-MSE we will compute the difference between metric X for MSE and metric X for BiVWAC (e.g. $\Delta_{Bias} = Bias_{MSE} - Bias_\alpha$). As we only care on which of them is larger than the others and not the actual range we will scale these values to [0,1]: $\Delta_X^{Scaled} = \frac{Bias_{MSE} - Bias_\alpha}{|Bias_{MSE}| + |Bias_\alpha|}$

In Figure 4 we can see that for BiVWAC-SAC, the reduction in MSE, variance and bias seem to correlate with the variation in performance of the agent: the lower metric $X_\alpha$ is compared to $X_{MSE}$, the better BiVWAC-$\alpha$ seems to perform. We also note that for Hopper, where AVEC ($\alpha = 0$) does not learn, the bias of AVEC is one to two orders of magnitude larger than SAC bias while the MSE and variance are still reduced from SAC.

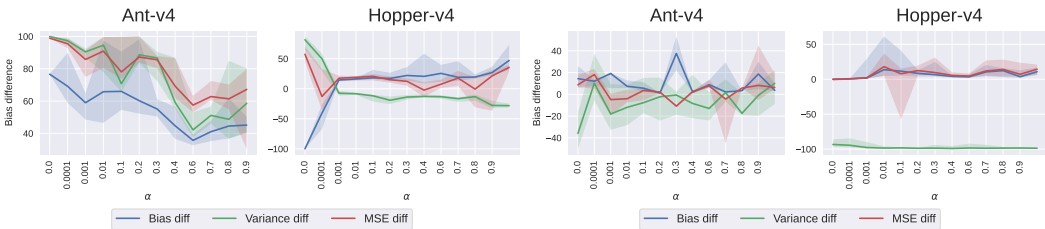

Figure 4: Scaled relative MSE difference, scaled relative bias difference and scaled relative variance difference over the last 100.00 training timesteps for Ant-v4 and over the 100.000 first timesteps for Hopper-v4 for BiVWAC-SAC (left) and BiVWAC-PPO (right) with different $\alpha$ values. Y-axis: Relative Metric Differences. X-axis: $\alpha$ values. Envelope represent one one standard deviation around the mean.

## 5 CONCLUSION

We introduced BiVWAC to control the bias-variance weighting in the critic objective through a hyperparamter $\alpha$. From this objective we can derive the MSE ($\alpha$= 0.5, weighting bias and variance equally) and AVEC ($\alpha = 0$, that only considers the variance). We demonstrated that BiVWAC is theoretically sound as it still leads to an unbiased estimation of the policy gradient. We motivated the need to extend beyond, or rather between, the MSE and AVEC and to study in-between values of $\alpha$ to find better weightings of bias and variance. We experimentally evaluated BiVWAC applied to two popular actor-critic algorithms SAC and PPO and have shown that the MSE is indeed a suboptimal weighting of bias and variance for critics. For BiVWAC-SAC, we found that $\alpha$ values close to 0 tend to provide better results than the MSE or AVEC. While it is possible to tune $\alpha$, we propose $\alpha^* = 10^{-4}$ as a safe value for which BiVWAC-SAC outperforms the MSE in all tasks and AVEC in almost all tasks (however AVEC fails to learn at all on other tasks). For BiVWAC-PPO we found that, while the pattern for the optimal value of $\alpha$ is harder to identify, there always exists values of $\alpha$ that perform better than the MSE, and in all tasks but one, values of $\alpha$ also performed better than AVEC. We measured the estimation and approximation error of the modified algorithms as well as the actor gradient estimation error in order to better understand the means through which BiVWAC impacts learning. As we intuited, we showed that BiVWAC, with the correct $\alpha$ values, leads to a better approximation of the true values through a reduction in variance, and that translates into a better estimation of the actor's gradient.

More work would be interesting to conduct regarding the relationship between $\alpha$ and the hyperparameters of the underlying algorithm, most importantly the one with a direct link to the estimations we impact: the batch size, the size of the neural network, the discount factor. Another interesting follow-up would be experimenting with schedules for $\alpha$, as the optimal-variance weighting may not be constant. Additionally, studying separate weightings for the two variances and the covariance in the MSE decomposition would allow for more flexibility at the expense of a larger hyperparameter space.

We conclude that, while BiVWAC-PPO still requires deeper analysis to propose heuristics regarding the choice of optimal $\alpha$ for a specific task, BiVWAC-SAC is a promising algorithm that should be extended to other tasks and algorithms to confirm the improvements shown in this work, and when used with $\alpha^* = 0.0001$ is an improvement over SAC in all studied tasks but one (13.5‰ average improvement) at virtually no additional cost and with minimal modifications.

### REPRODUCIBILITY

The Mujoco tasks were accessed through the Gymnasium (Towers et al., 2024) library. We use the stable-baselines3 (Raffin et al., 2021) implementations of SAC and PPO for ease of comparison with other works. We modify these implementations following the BiVWAC scheme as well as adding logging for metrics to analyze the behavior of the agents. Using a popular implementation we aim to show that the performance variation originate from BiVWAC and not from other differences in implementation. Hyperparameters used were taken from rl-zoo (Raffin, 2020) and are optimized for each environment. For more details see Appendix F).

The seeds used are the sequence of the first $n$ non-negative integers starting with 0, with $n$ the number of required seeds.

We want to put emphasis on the number of experiments needed to reproduce the results of this work. The number of experiments allows for more statistically robust results and more tasks to test our method on for more hyperparameter values, which should enhance the ease of reproducing our results. However this also makes the results harder to replicate without access to the proper computing power.

The code to reproduce the experiments and regenerate the plots is available at `https://anonymous.4open.science/r/AVEC-D11A/`.

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

## A    APPENDIX: MSE BIAS VARIANCE DECOMPOSITION

### A.1    PROOF OF LEMMA 2.1

We have:

$$
\begin{aligned}
MSE(\hat{y}, y) &= \mathbb{E}\left[(\hat{y} - y)^2\right] \\
&= \mathbb{E}\left[(\hat{y} - y - \mathbb{E}[\hat{y} - y] + \mathbb{E}[\hat{y} - y])^2\right] \\
&\overset{(a)}{=} \mathbb{E}\left[(\hat{y} - y - \mathbb{E}[\hat{y} - y])^2\right] + \mathbb{E}[\hat{y} - y]^2 + 2\mathbb{E}[\hat{y} - y]\mathbb{E}\left[\hat{y} - y - \mathbb{E}[\hat{y} - y]\right] \\
&\overset{(b)}{=} \mathbb{E}\left[(\hat{y} - y - \mathbb{E}[\hat{y} - y])^2\right] + \mathbb{E}[\hat{y} - y]^2 \\
&= \mathrm{Var}\left[\hat{y} - y\right] + \mathbb{E}[\hat{y} - y]^2
\end{aligned}
$$

Where $(a)$ follows from expansion of the square and $(b)$ by linearity of expectation showing that the last term is zero. Then, because the covariance is bilinear,

$$
MSE(\hat{y}, y) = \mathrm{Var}\left[\hat{y}\right] + \mathrm{Var}\left[y\right] - 2Covar(y, \hat{y}) + \mathrm{Bias}(\hat{y}, y)^2.
$$

### A.2    WEIGHTED MSE EQUALITY, PROOF OF LEMMA 3.1

Let us show that
$$
MSE_\alpha(z - \hat{z}, 0) = MSE_\alpha(z, \hat{z}). \tag{9}
$$
We have from equation 5 applied to $\hat{y} = z - \hat{z}$ and $y = 0$,

$$
\begin{aligned}
MSE_\alpha(z - \hat{z}, 0) &= \alpha \,\mathrm{Bias}(z - \hat{z}, 0)^2 + (1 - \alpha)\left(\mathrm{Var}(z - \hat{z}) - 2Covar(z - \hat{z}, 0) + \mathrm{Var}(0)\right) \\
&= (1 - \alpha)\,\mathrm{Var}(z - \hat{z}) + \alpha \,\mathrm{Bias}(z - \hat{z}, 0)^2
\end{aligned}
$$

using that $Covar(z - \hat{z}, 0) = \mathbb{E}[(z - \hat{z} - \mathbb{E}[z - \hat{z}])0] = 0$ and $\mathrm{Var}(0) = 0$. Then, having

$$
\mathrm{Bias}(z - \hat{z}, 0) = \mathbb{E}\left[\mathbb{E}\left[z - \hat{z}\right] - 0\right] = \mathbb{E}\left[z - \hat{z}\right] = \mathrm{Bias}(z, \hat{z}),
$$

we get

$$
\begin{aligned}
MSE_\alpha(z - \hat{z}, 0) &= (1 - \alpha)\,\mathrm{Var}(z - \hat{z}) + \alpha\,\mathrm{Bias}(z, \hat{z})^2 \\
&\overset{(a)}{=} (1 - \alpha)\left(\mathrm{Var}(z) + \mathrm{Var}(\hat{z}) - 2Covar(z, \hat{z})\right) + \alpha\,\mathrm{Bias}(z, \hat{z})^2 \\
&= MSE_\alpha(z - \hat{z}, 0),
\end{aligned}
$$

where the last equality follows from the definition of $MSE_\alpha$ (equation 5) and $(a)$ follows by bilinearity of the covariance.

## B    APPENDIX: UNBIASED BIAS-VARIANCE-WEIGHTED ACTOR-CRITIC: PROOF OF THEOREM 3.2

We want to show that there exists $g_\phi$ a function from $\mathcal{S} \times \mathcal{A}$ to $\mathbb{R}$ that depends only on $\phi$ such that

$$
\nabla_\theta J = \mathbb{E}_{(s,a) \sim \pi}\left[\nabla_\theta \log(\pi_\theta) g_\phi(s, a)\right].
$$

We will do the proof for the $Q(s_t, a_t)$ version, but it also holds for the $V(s_t)$.
First we compute the gradient of the BiVWAC-loss, let's start by recalling its expression (see equation 6)

$$
\mathcal{L}_{\mathrm{BiVWAC}} = (1 - \alpha)\,\mathrm{Var}(f_\phi(s, a) - \hat{Q}^\pi(s, a)) + \alpha.\,\mathrm{Bias}^2(f_\phi(s, a), \hat{Q}^\pi(s, a)) \text{ with } \alpha \in [0, 1). \tag{10}
$$

We decompose the computation of the gradient of the loss in two terms: $\nabla_\phi \mathrm{Var}$ and $\nabla_\phi \mathrm{Bias}^2$.

**Step 1: Computation of $\nabla_\phi \text{Var}$.** We have,

$$\text{Var}(f_\phi(s,a) - \hat{Q}^\pi(s,a)) = \mathbb{E}_{(s,a)\sim\pi}[(f_\phi(s,a) - \hat{Q}^\pi(s,a))^2] - \mathbb{E}_{(s,a)\sim\pi}[(f_\phi(s,a) - \hat{Q}^\pi(s,a))]^2$$

Then, taking the gradient, because $\hat{Q}^\pi(s,a)$ does not depend on $\phi$,

$$\nabla_\phi \text{Var}(f_\phi(s,a) - \hat{Q}^\pi(s,a)) = 2\mathbb{E}_{(s,a)\sim\pi}\Bigg[(f_\phi(s,a) - \hat{Q}^\pi(s,a) - \mathbb{E}_{(s,a)\sim\pi}[f_\phi(s,a) - \hat{Q}^\pi(s,a)])$$

$$\left(\nabla_\phi f_\phi(s,a) - \mathbb{E}_{(s,a)\sim\pi}\left[\nabla_\phi f_\phi(s,a)\right]\right)\Bigg]$$

then, because the left term in brackets is centered we have:

$$\nabla_\phi \text{Var} = 2(\mathbb{E}_{(s,a)\sim\pi}\left[(f_\phi(s,a) - \hat{Q}^\pi(s,a) - \mathbb{E}_{(s,a)\sim\pi}[f_\phi(s,a) - \hat{Q}^\pi(s,a)])\nabla_\phi f_\phi(s,a)\right]$$

$$- \underbrace{\mathbb{E}_{(s,a)\sim\pi}\left[f_\phi(s,a) - \hat{Q}^\pi(s,a) - \mathbb{E}_{(s,a)\sim\pi}\left[f_\phi - \hat{Q}^\pi(s,a)\right]\right]}_{0}\mathbb{E}_{(s,a)\sim\pi}[\nabla_\phi f_\phi(s,a)]$$

$$= 2\mathbb{E}_{(s,a)\sim\pi}\left[(f_\phi(s,a) - \hat{Q}^\pi(s,a) - \mathbb{E}_{(s,a)\sim\pi}[f_\phi(s,a) - \hat{Q}^\pi(s,a)])\nabla_\phi f_\phi(s,a)\right]. \quad (11)$$

**Step 2: Computation of $\nabla_\phi \text{Bias}^2$.** Taking the gradient in the bias, we get,

$$\text{Bias}^2 = \mathbb{E}_{(s,a)\sim\pi}[f_\phi(s,a) - \hat{Q}^\pi(s,a)]^2$$

$$\nabla_\phi \text{Bias}^2 = 2\mathbb{E}_{(s,a)\sim\pi}\left[\nabla_\phi f_\phi(s,a)\mathbb{E}_{(s,a)\sim\pi}[f_\phi(s,a) - \hat{Q}^\pi(s,a)]\right]. \quad (12)$$

**Step 3: Computation of $\nabla_\phi \mathcal{L}_{\text{BiVWAC}}$.** We can now inject derivatives of $\nabla_\phi \text{Var}$ and $\nabla_\phi \text{Bias}^2$ from equation 11 and equation 12 into the loss equation 10, we get

$$\nabla_\phi \mathcal{L}_{\text{BiVWAC}} = 2(1-\alpha)\mathbb{E}_{(s,a)\sim\pi}\left[(f_\phi(s,a) - \hat{Q}^\pi(s,a) - \mathbb{E}_{(s,a)\sim\pi}[f_\phi(s,a) - \hat{Q}^\pi(s,a)])\nabla_\phi f_\phi\right]$$

$$+ 2\alpha\mathbb{E}_{(s,a)\sim\pi}\left[\nabla_\phi f_\phi\mathbb{E}_{(s,a)\sim\pi}[f_\phi(s,a) - \hat{Q}^\pi(s,a)]\right]$$

$$= 2(1-\alpha)\mathbb{E}_{(s,a)\sim\pi}\left[(f_\phi(s,a) - \hat{Q}^\pi(s,a))\nabla_\phi f_\phi(s,a)\right]$$

$$+ 2(2\alpha - 1)\mathbb{E}_{(s,a)\sim\pi}\left[\mathbb{E}_{(s,a)\sim\pi}\left[f_\phi(s,a) - \hat{Q}^\pi(s,a)\right]\nabla_\phi f_\phi(s,a)\right].$$

**Step 4: re-expression of $\nabla_\phi \mathcal{L}_{\text{BiVWAC}}$ using the policy parameterization assumption and expression for $g_\phi$.** Under the policy parameterization assumption ((Sutton & Barto, 2020, parametrization assumption: 13.1, Policy gradient theorem : Section 13.2)) we have $\nabla_\phi f_\phi = \nabla_\theta \log(\pi_\theta)$ and $\nabla_\theta J = \mathbb{E}[Q\nabla_\theta \log \pi_\theta]$, hence,

$$\frac{\nabla_\phi \mathcal{L}_{\text{BiVWAC}}}{2} = (1-\alpha)(\nabla_\theta J - \mathbb{E}_{(s,a)\sim\pi}\left[f_\phi(s,a)\nabla_\theta \log(\pi_\theta)\right]$$

$$+ (2\alpha - 1)\mathbb{E}_{(s,a)\sim\pi}\left[\mathbb{E}_{(s,a)\sim\pi}\left[f_\phi(s,a) - \hat{Q}^\pi(s,a)\right]\nabla_\theta \log(\pi_\theta)\right].$$

When a local optimum is reached, the gradient of the loss is zero, thus we have:

$$0 = (1-\alpha)(\nabla_\theta J - \mathbb{E}_{(s,a)\sim\pi}\left[f_\phi(s,a)\nabla_\theta \log(\pi_\theta)\right] + (2\alpha - 1)\mathbb{E}_{(s,a)\sim\pi}\left[\mathbb{E}_{(s,a)\sim\pi}\left[f_\phi(s,a) - \hat{Q}^\pi(s,a)\right]\nabla_\theta \log(\pi_\theta)\right].$$

Hence, if we isolate the expression of $\nabla_\theta J$ and using that $\nabla_\theta \log(\pi_\theta)$ does not depend on the action stats $(s,a)$, we get

$$\nabla_\theta J = \mathbb{E}_{(s,a)\sim\pi}\left[f_\phi(s,a)\nabla_\theta \log(\pi_\theta)\right] - \frac{2\alpha - 1}{1-\alpha}\mathbb{E}_{(s,a)\sim\pi}\left[f_\phi(s,a) - \hat{Q}^\pi(s,a)\right]\nabla_\theta \log(\pi_\theta)$$

$$= \mathbb{E}_{(s,a)\sim\pi}\left[\nabla_\theta \log(\pi_\theta)\left(f_\phi(s,a) + \frac{1 - 2\alpha}{1-\alpha}\mathbb{E}_{(s,a)\sim\pi}\left[f_\phi(s,a) - \hat{Q}^\pi(s,a)\right]\right)\right].$$

We can finally identify the form of $g_\phi$ and prove the intended result

$$\nabla_\theta J = \mathbb{E}_{(s,a)\sim\pi}\left[\nabla_\theta \log(\pi_\theta)g_\phi(s,a)\right], \text{ with } g_\phi(s,a) = f_\phi(s,a) + \frac{1-2\alpha}{1-\alpha}\mathbb{E}_{(s,a)\sim\pi}\left[f_\phi(s,a) - \hat{Q}^\pi(s,a)\right].$$

## C    BiVWAC-PPO

---

**Algorithm 2** BiVWAC-PPO

---

**Input parameters** : $\lambda_\pi \geq 0, \lambda_V \geq 0, 0 \leq \alpha < 1$
**Initialize** policy parameters $\theta$ and value-function parameters $\phi$
**for** each update step **do**
    batch $\mathcal{B} \leftarrow \emptyset$
    **for** each environment step **do**
        $a_t \sim \pi_\theta(s_t)$
        $s_{t+1} \sim \mathcal{P}(s_t, a_t)$
        $\mathcal{B} \leftarrow \mathcal{B} \cup \{(s_t, a_t, r_t, s_{t+1})\}$
    **end for**
    $A \leftarrow 0^{|\mathcal{B}|} \in \mathbb{R}^{|\mathcal{B}|}$
    $\lambda_{gae} \leftarrow 0$
    $\delta_{\text{BiVWAC}} \leftarrow \frac{1-2\alpha}{1-\alpha} \cdot \frac{1}{|\mathcal{B}|} \cdot \sum_{t=0}^{|\mathcal{B}|}(R_t - f_\phi(s_t))$         $\triangleright$ Estimate the correction term
    **for** each transition in $\mathcal{B}$ in **reverse do**         $\triangleright$ GAE using $g_\phi$ instead of $\hat{V}^\pi$
        $g_\phi \leftarrow f_\phi + \delta_{\text{BiVWAC}}$
        $\Delta \leftarrow r_t + \gamma g_\phi(s_{t+1}) - g_\phi(s_t)$         $\triangleright$ Use $g_\phi$ instead of $f_\phi$
        $\lambda_{gae} \leftarrow \Delta + \gamma \lambda_{gae}$
        $A_t \leftarrow \lambda_{gae}$
    **end for**
    **for** each gradient step **do**
        $J^{PPO} = function(A)$         $\triangleright$ Replace advantages with corrected advantages in loss
        $\theta \leftarrow \theta - \lambda_\pi \hat{\nabla}_\theta J^{PPO}$
        $\phi \leftarrow \phi - \lambda_V \hat{\nabla}_\phi \mathcal{L}_{\text{BiVWAC}}$
    **end for**
**end for**
Differences between BiVWAC-PPO and PPO are highlighted on the right side.

---

# D MEANINGFUL CRITIC COMPARISONS

We begin by stating that had we trained two separate instances of the same algorithm with only the critic loss changing we could not have compared their critic's performance as it would have been trained using different data. This would lead to problems such as:

- The states encountered depend on the policy collecting them. As a consequence, the critic's performance depends on how hard the values of states encountered are to estimate. One critic leading to lesser actor performance may stagnate collecting low rewards leading to a seemingly easier task than another critic leading to good actor performance but encountering new and harder to estimate states. This leads to the impossibility of comparing the critics with each observing data collected from a different policy, they need to use the same data.
- As the states encountered differ, one would have to decide which states would be used to evaluate algorithms, so that they do not favor one or the other critic. Finding such balance seems like a difficult task as regression models tend to perform unpredictably on data too different from what the one they were trained on and thus adding data seen by one critic would lead to possibly non-informative results for the other critic, and using data not seen by both would lead to even less informative results. If the two algorithms had collected data from states suffienctly similar, this approach would work. However that would seriously limit the range of algorithms this method can compare.

To tackle these issues, we propose the following methodology to compare the same algorithm with two different critics:

1. We initialize two critics $C_1$ and $C_2$ sharing the same parameters at initialization.
2. $C_1$ is trained using $\mathcal{L}_{\text{BiVWAC}}$, our critic loss. It is then used to compute the actor loss. The actor is then used to select actions.
3. Meanwhile, $C_2$ is also trained on the same data as $C_1$, however it has no impact on the actor, and thus data collection.
4. During training, as often as needed, we measure the value estimations $f_\phi^{C_1} and f_\phi^{C_2}$ by using $C_1$ and $C_2$ on the same states $\mathcal{S}_{eval}$ sampled from states recently seen by the agent. We also collect their corresponding surrogate targets $\hat{Q}_{C_1}^{\pi_\theta} and \hat{Q}_{C_2}^{\pi_\theta}$ (the same is valid for $V^{\pi_\theta}$).
5. After training we estimate the true values of $\mathcal{S}_{eval}$ using Monte-Carlo (MC) rollouts.
6. We compare the variance, Estimation error and Approximation error of $C_1$ and $C_2$ using the collected values.

For every 10% of the total number of timesteps collected during training, we experimentally approximate the true value of the policy. We approximate it on 50 states randomly sampled from the current agent's buffer (50 states from the rollout buffer for PPO and 50 states from the states of the Replay buffer added since the last evaluation for SAC, e.g. from the last 1000 states added in the buffer if the number of timesteps was 10,000). The true values are then estimated by collecting $5\dot{1}0^5$ samples from these states (and a given action for SAC as we estimate $Q^\pi(s, a)$) using the current policy and computing the average discounted-return through Monte-Carlo estimation.

# E ADDITIONAL FIGURES

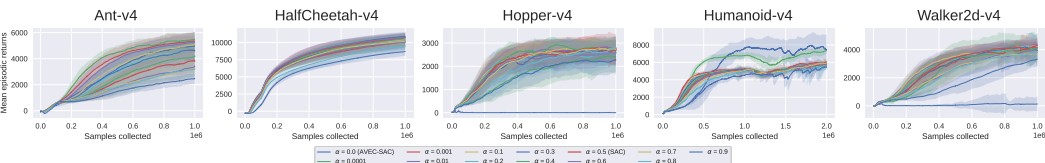

Figure 5: Episodic mean cumulative rewards of BiVWAC-SAC for different $\alpha$ values on MuJoCo tasks. X-axis: number of collected samples. Y-axis: episodic cumulative reward.

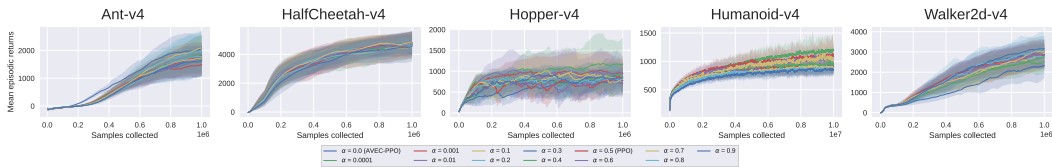

Figure 6: Episodic mean cumulative rewards of BiVWAC-PPO for different $\alpha$ values on MuJoCo tasks. X-axis: number of collected samples. Y-axis: episodic cumulative reward.

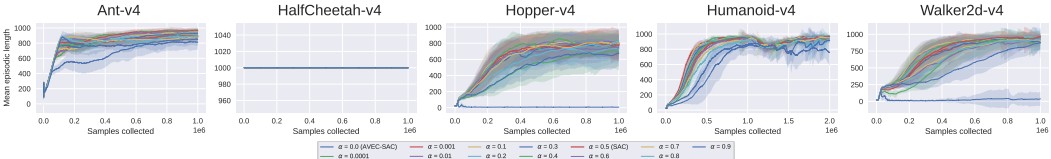

Figure 7: Episodic mean length of BiVWAC-SAC for different $\alpha$ values on MuJoCo tasks. X-axis: number of collected samples. Y-axis: episodic length.

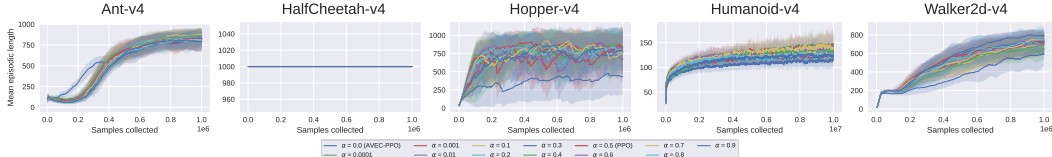

Figure 8: Episodic mean length of BiVWAC-PPO for different $\alpha$ values on MuJoCo tasks. X-axis: number of collected samples. Y-axis: episodic length.

# F EXPERIMENTAL DETAILS

Experiments were made using stable-baselines3's implementations of SAC and PPO and applying modifications for the BiVWAC loss and the logging of the metrics reported in the paper.

## F.1 IMPLEMENTATION DETAILS

Because we only use ADAM for optimizing, and because it is scale invariant, we do not need to scale the loss to its original size when using $\alpha = 0.5$ to get back the original MSE loss for the critic.

For true value estimation, we collect $5\dot{1}0^5$ samples starting from the given state and using the current policy. To speed up the process we collect it from multiple environments at the same time (32 environments). As we use Monte-Carlo estimation of Returns, we can only consider full episodes. As a consequence, we discard the last, unfinished episode, for each parallel environment. Below are represented the actual number of complete episodes and timesteps considered for each experiment:

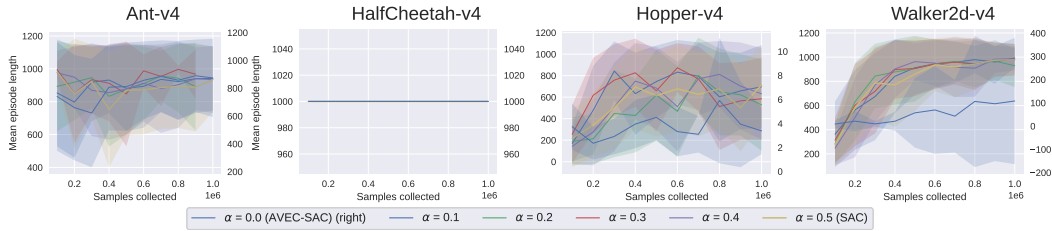

Figure 9: Mean over 32 envs of Episodic mean length of episodes collected during MC rollouts. Envelope represents a standard deviation from the mean). X-axis: number of collected samples. Y-axis: episodic length.

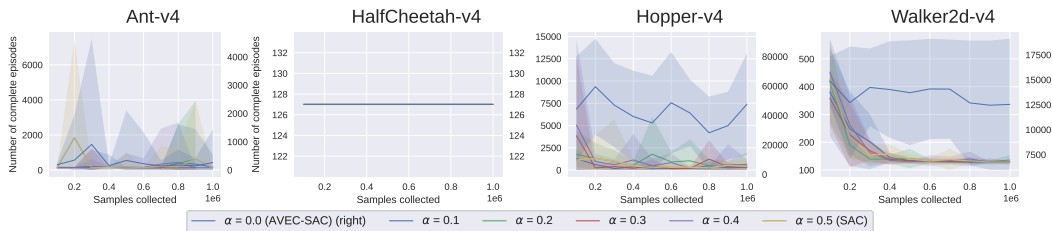

Figure 10: Mean number of episodes collected per parallel environment during MC rollouts. Envelope represents a standard deviation from the mean). X-axis: number of collected samples. Y-axis: episodic length.

## F.2 HYPERPARAMETERS

In Tables 2 and 4 we report the list of hyperparameters common to all continuous control experiments. All environments use the default hyperparameters unless specified otherwise in Table 3, 5, and 6. Hyperparameters are taken from stable-baselines3 (Raffin et al., 2021) default parameters, and rl-zoo3 (Raffin, 2020) when they differ from the default stable-baselines3 value. rl-zoo3 provides optimized hyperparemeters for different agents and environments. Note that the parameters were optimized for the original algorithms (PPO and SAC) and not for they modified versions (BiVWAC-PPO and BiVWAC-SAC). Hence they should favor the original algorithms.

Table 2: Default hyperparameters for both SAC and BiVWAC-SAC.

| Parameter | Value |
|---|---|
| Number of training steps | $10^6$ |
| Adam stepsize | $3 \cdot 10^{-4}$ |
| Discount ($\gamma$) | 0.99 |
| Replay buffer size | $10^6$ |
| Batch size | 256 |
| Nb. hidden layers | 2 |
| Nb. hidden units per layer | 256 |
| Nonlinearity | ReLU |
| Target smoothing coefficient ($\tau$) | 0.005 |
| Target update interval | 1 |
| Gradient steps | 1 |
| Learning starts | $10^4$ |

Table 3: Environment specific hyperparameters for SAC and BiVWAC-SAC.

| Environment | Parameter | Value |
|---|---|---|
| Humanoid-v4 | Number of training steps | $2 \cdot 10^6$ |

Table 4: Default hyperparameters for both PPO and BiVWAC-PPO.

| Parameter | Value |
|---|---|
| Number of training steps | $10^6$ |
| Horizon ($T$) | 2048 |
| Adam stepsize | $3 \cdot 10^{-4}$ |
| Batch size | 64 |
| Nb. epochs | 10 |
| Nb. hidden layers | 2 |
| Nb. hidden units per layer | 64 |
| Nonlinearity | tanh |
| Discount ($\gamma$) | 0.99 |
| GAE parameter ($\lambda$) | 0.95 |
| Clipping parameter ($\epsilon$) | 0.2 |
| Maximum gradient norm | 0.5 |
| State and reward normalization | True |
| Nb. environments | 1 |
| Value function loss coefficient | 0.5 |
| Initialization log standard deviation | 0.0 |
| Orthogonal initialization | True |

Table 5: Environment specific hyperparameters for PPO and BiVWAC-PPO.

| Environment | Parameter | Value |
|---|---|---|
| HalfCheetah-v4 | Discount ($\gamma$) | 0.98 |
| | Horizon ($T$) | 512 |
| | Adam stepsize | $2.0633 \cdot 10^{-5}$ |
| | Entropy coeffecient | 0.000401762 |
| | Clipping parameter ($\epsilon$) | 0.1 |
| | Nb. epochs | 20 |
| | GAE parameter ($\lambda$) | 0.92 |
| | Maximum gradient norm | 0.8 |
| | Value function loss coefficient | 0.58096 |
| | Initialization log standard deviation | -2.0 |
| | Orthogonal initialization | False |
| | Nb. hidden units per layer | 256 |
| | Nonlinearity | ReLU |
| | | |
| Hopper-v4 | Discount ($\gamma$) | 0.999 |
| | Horizon ($T$) | 512 |
| | Adam stepsize | $9.80828 \cdot 10^{-5}$ |
| | Batch size | 32 |
| | Entropy coeffecient | 0.00229519 |
| | Nb. epochs | 5 |
| | GAE parameter ($\lambda$) | 0.99 |
| | Maximum gradient norm | 0.7 |
| | Value function loss coefficient | 0.835671 |
| | Initialization log standard deviation | -2.0 |
| | Orthogonal initialization | False |
| | Nb. hidden units per layer | 256 |
| | Nonlinearity | ReLU |
| | | |
| Humanoid-v4 | Number of training steps | $10^7$ |
| | Discount ($\gamma$) | 0.95 |
| | Horizon ($T$) | 512 |
| | Adam stepsize | $3.56987 \cdot 10^{-5}$ |
| | Batch size | 256 |
| | Entropy coeffecient | 0.00229519 |
| | Clipping parameter ($\epsilon$) | 0.3 |
| | Nb. epochs | 5 |
| | GAE parameter ($\lambda$) | 0.9 |
| | Maximum gradient norm | 2.0 |
| | Value function loss coefficient | 0.431892 |
| | Initialization log standard deviation | -2.0 |
| | Orthogonal initialization | False |
| | Nb. hidden units per layer | 256 |
| | Nonlinearity | ReLU |

Table 6: Environment specific hyperparameters for PPO and BiVWAC-PPO.

| Environment | Parameter | Value |
|---|---|---|
| Walker2d-v4 | Discount ($\gamma$) | 0.99 |
| | Horizon ($T$) | 512 |
| | Adam stepsize | $5.05041 \cdot 10^{-5}$ |
| | Batch size | 32 |
| | Entropy coefficent | 0.00229519 |
| | Clipping parameter ($\epsilon$) | 0.1 |
| | Nb. epochs | 20 |
| | GAE parameter ($\lambda$) | 0.95 |
| | Maximum gradient norm | 1 |
| | Value function loss coefficient | 0.871923 |

