# OpenReview forum: "BiVWAC: Improving deep reinforcement learning algorithms using Bias-Variance Weighted Actor-Critic"
_ICLR.cc/2025/Conference — ICLR 2025 Conference Withdrawn Submission_

### Official Review · Reviewer_usno · 2024-10-31

**Soundness:** 3
**Presentation:** 2
**Contribution:** 2
**Rating:** 3
**Confidence:** 4

**Summary:**

This paper extend the AVEC framework by introducing a hyperparameter $\alpha$ to control the balance between bias and variance.

**Strengths:**

1. This paper introduce a novel $\alpha$ to balance bias and variance.
2. This papre provide some theoretical analysis.

**Weaknesses:**

1. This paper addresses the issue of bias between the function approximation $\bar{Q^\pi}$ and its target $y$. My understanding is that this approach aims to mitigate the challenge that the true value of the function is unknown. However, as shown in [1], the TD error is an inadequate substitute for the true value error, which can significantly impact RL performance. Although controlling the bias between the function approximation and its target is possible, the value error may still remain substantial. Could the author provide some theoretical analysis on this value error aspect?
2. Although the $\alpha$ is a novel contribution, the result does not reflect on this. Based on Figure 1, for SAC, $\alpha = 0$ or $\alpha$ close to 0 result in best performance. However if the $\alpha = 0$, based on Equation 6, the loss function is exactly AVEC cost.



[1] Fujimoto, Scott, et al. "Why should i trust you, bellman? the bellman error is a poor replacement for value error." arXiv preprint arXiv:2201.12417 (2022).

**Questions:**

1. For Table 1, auther did not report which $\alpha$ used for the evaluation. Do you use the same $\alpha$ for all envs or you use different?
2. For Table 1, based on AVEC paper, the walker2d, AVEC-SAC is $4334 \pm 128$ however in this paper, AVEC-SAC is $119 \pm 488$. Additionally, the std value for Table 1 are all greater than AVEC paper. This may because of the gym version difference. However to avoid confusion and concerns, can auther perform another experiment using the same gym version as AVEC paper? (Just Walker2d is sufficent for the rebuttal)
3. On page 9, line 478, there is a placeholder indicating "Appendix ??," which appears to be a missing reference.
4. In Figure 4, for SAC, the $\alpha \to 0$ result in larger bias and variance but in Figure 1, $\alpha \to 0$ result in best performance. Why controlling bias and variance does not inprove performance?

---

### Official Review · Reviewer_mzxp · 2024-11-02

**Soundness:** 2
**Presentation:** 2
**Contribution:** 2
**Rating:** 3
**Confidence:** 3

**Summary:**

This paper proposes an algorithm in which the bias and the variance components of the mean squared error in temporal difference learning can be traded off in a flexbible way. For this, the authors derive a loss function based on the convex combination of the bias-variance terms in the MSE. The authors conduct experiments on the Mujoco suite for various interpolation values and show a strong dependency on this parameter. Consequently, the authors suggest that appropriate tuning of this trade-off parameter can benefit learning. The authors moreover provide a reference value that works reasonably well on a range of environments.

**Strengths:**

- The paper investigates an interesting issue that is not well-understood in the community, especially when TD-learning is combined with neural function approximation.
- The proposed method, from what I can see, is applicable to a wide range of algorithms, as the number of algorithms emloying the MSE in TD learning is large.
- The experiments involve a relatively large amount of seeds and involve a wide range of values o
fr $\alpha$.

**Weaknesses:**

- To me, the main weakness of this paper in its current form is its presentation. Despite being a fairly straightforward reformulation of the mean squared error, I found the motivation and derivations hard to follow. For example, the equations in lines 64, 144, 180, 190, 229, and 246 all serve to arrive at the loss given in eq. 6 but in a rather cumbersome way, introducing several variables and notation that are not defined explicitly. I suggest that the derivation of Eq. 6 could be significantly more concise by showing that $\mathcal{L}_{avec}$ is in fact the variance term of a bias-variance decomposition of the MSE and that one can recover the MSE by adding to this the bias term with equal weighting.

- Conceptually, I am unsure whether this approach should be thought of as an "interpolation" between AVEC and MSE in the sense that AVEC is a special case of $\alpha=0$. There is a significant difference, in my opinion, between weighing the components in line 260 and 262 differently or having $\alpha=0$ (AVEC) and $\alpha=1$. This is because AVEC changes the minimizer of the objective function, whereas the shown approach mainly changes the weighting of gradients. For example, AVEC is not a sensible algorithm without the correction term, wheres one can argue that any value of $\alpha>0$ and $\alpha<1$ still shares the same minimizer as
the MSE (assuming sufficient expressivity and continuous state-spaces).

- The experimental results seem to speak to the above points, in that $\alpha=0$ is a significantly different algorithm. There are moreover a few experimental results that I find concerning:
   - In Fig. 3, several versions with $\alpha=0$ (AVEC) perform worse with the gradient correction. In my mind this is a highly counterintuitive result. In my understanding, the objective function of AVEC has no reason to provide accurate gradients without the correction term, so it is highly surprising to see higher performance without it.
   - I find Fig. 4 a very difficult to interpret plot. For example, I don't follow the authors suggestion that the trend of the curve is meaningful. The sign of the shown curves seems like a more relevant metric: For example, all curves on Ant-v4 l.h.s. seem to indicate that bias, variance, and MSE are reduced in the alpha version. This, however, does not really corroborate the trade-off nature suggested by the authors. But what I find more concerning is that I expected all curves to cross 0 at alpha=0.5, as the authors suggest that the loss equals the MSE for this value. Most curves, however, stray significantly far away from 0. The above points, in my opinion, cast serious doubt as to whether the bias-variance trade-off is the driving factor in the observed experimental results.

- The related work section seems rather short. Several classical works (e.g. many works by Sutton, Singh, Bertsekas) discuss the bias-variance tradeoff at length in temporal difference methods, policy gradients, etc.

**Questions:**

- Why are the experiments in Fig. 3 l.h.s. limited to three environments?
- How significant is the batch size in computing the empirical bias terms?

---

### Official Review · Reviewer_8247 · 2024-11-03

**Soundness:** 3
**Presentation:** 1
**Contribution:** 2
**Rating:** 3
**Confidence:** 4

**Summary:**

This paper proposes BiVWAC, a weighted sum of standard MSE and AVEC (Actor with Variance Estimated Critic) loss, which is derived by applying bias-variance decomposition to critic's residual error. It is experimentally shown that BiVWAC improves the performance of SAC and PPO, if the weight of MSE and AVEC losses are appropriately chosen. It is also shown that, an unbiased policy gradient estimator is also constructed for critics learned from BiVWAC loss, though the experimental results indicated that the uncorrected biased estimators likely perform well in practice.

**Strengths:**

- Bias-variance trade-off is a fundamental topic in learning theories. In particular, variance reduction techniques are also important methods in policy gradient based reinforcement learning methods. Theoretical analyses and practical algorithms in this direction have a large group of potential audiences (significance).
- Theoretical results look correct and sound (quality).
- Experimental results indicate the potential effectiveness of variance reduction methods based on bias-variance decomposition (quality, significance).

**Weaknesses:**

- Inconsistency of the theory and the experiment.
  - Throughout the exposition of the theoretical arguments, the "true" target is un-regularized value $Q^{\pi}$ (or $V^{\pi}$). Theorem 3 also states that $g_\phi$ can be used to construct an unbiased estimate of conventional un-regularized policy gradient.
  - On the other hand, the base algorithms in the experiments are SAC and PPO, both of which do not align these un-regularized argument.
    - For SAC, the true target is the entropy regularized value, $Q_{\tau}^{\pi} = R(s,a) + \gamma \sum_{s',a'} P(s'|s,a)\pi(a'|s') (Q_{\tau}^{\pi}(s',a') - \tau \log \pi(a'|s'))$. In addition, the actor loss is a KL divergence between the parameterized policy and the energy based policy induced by the regularized value.
    - For PPO, the actor loss incorporates clipped log-ratio values to implicitly apply trust-region constraint.
  - Therefore, the theoretical results obtained in this paper does not explain the experimental behaviors of BiVWAC-SAC and BiVWAC-PPO.

- A rather minor concern is that, the quality of writing seems not satisfactory.
  - The expositions are not consistent in some parts.
    - Section 2.2, the author explored to which quantities Lemma 2.1 should be applied, and states that "As the policy gradient $\nabla_{\theta} J$ directly reflects our objective of maximizing $J$, it is the best candidate". However, Lemma 2.1 is not directly applied to $\nabla J $ but to the critic's residual error.
    - In L.118-119, it is stated that "In this work we limit our scope to policies which can be represented by Gaussian distributions". However, I found no gaussian requirements in the theoretical arguments.
  - Followings are the flaws in the writing that I noticed.
    - Italic and normal characters are mixed up for MSE, Bias and Var. it is recommended to use either of them consistently.
    - L.239/240: to be to be
    - the sentence after Eq. (8) lacks a period.
    - L.318: $\delta_{\rm BiVWAC}$ is used without clear definition.
    - L.414: Figure 2 -> Figure 1?
    - L.685; last one must be MSE_aloha(z,z-)

**Questions:**

- I suggest to revise the whole manuscript and check the consistency of the arguments and the provided theoretical/experimental results.

---

### Note · Authors · 2024-11-21

**Comment:**

Dear reviewers and chairs,

Thanks for your time and consideration. After carefully reading your reviews we decided to withdraw our submission to revise our manuscript.

Kind regards,

The authors

**Withdrawal Confirmation:**

I have read and agree with the venue's withdrawal policy on behalf of myself and my co-authors.